# Different Alterations of Hippocampal and Reticulo-Thalamic GABAergic Parvalbumin-Expressing Interneurons Underlie Different States of Unconsciousness

**DOI:** 10.3390/ijms24076769

**Published:** 2023-04-05

**Authors:** Ljiljana Radovanovic, Andrea Novakovic, Jelena Petrovic, Jasna Saponjic

**Affiliations:** Institute of Biological Research “Sinisa Stankovic”, National Institute of the Republic of Serbia, University of Belgrade, 11060 Belgrade, Serbia

**Keywords:** GABAergic parvalbumin-expressing interneurons, unconsciousness, sleep, anesthesia, hippocampus, reticulo-thalamic nucleus, postsynaptic density protein 95 (PSD-95), EEG microstructure

## Abstract

We traced the changes in GABAergic parvalbumin (PV)-expressing interneurons of the hippocampus and reticulo-thalamic nucleus (RT) as possible underlying mechanisms of the different local cortical and hippocampal electroencephalographic (EEG) microstructures during the non-rapid-eye movement (NREM) sleep compared with anesthesia-induced unconsciousness by two anesthetics with different main mechanisms of action (ketamine/diazepam versus propofol). After 3 h of recording their sleep, the rats were divided into two experimental groups: one half received ketamine/diazepam anesthesia and the other half received propofol anesthesia. We simultaneously recorded the EEG of the motor cortex and hippocampus during sleep and during 1 h of surgical anesthesia. We performed immunohistochemistry and analyzed the PV and postsynaptic density protein 95 (PSD-95) expression. PV suppression in the hippocampus and at RT underlies the global theta amplitude attenuation and hippocampal gamma augmentation that is a unique feature of ketamine-induced versus propofol-induced unconsciousness and NREM sleep. While PV suppression resulted in an increase in hippocampal PSD-95 expression, there was no imbalance between inhibition and excitation during ketamine/diazepam anesthesia compared with propofol anesthesia in RT. This increased excitation could be a consequence of a lower GABA interneuronal activity and an additional mechanism underlying the unique local EEG microstructure in the hippocampus during ketamine/diazepam anesthesia.

## 1. Introduction

The anesthetic state and natural sleep share many neurobiological features, yet they are distinct states [1,2,3,4]. While the anesthetic state is a pharmacologically induced, reversible state of unconsciousness, sleep is endogenously generated and involves the active suppression of consciousness by nuclei in the brainstem, diencephalon, and basal forebrain, and is dependent on homeostatic drive and circadian rhythm [5]. In contrast with sleep, in which there is a regular switch between non-rapid eye movement (NREM) and rapid eye movement (REM) sleep, once NREM sleep is established, in anesthesia at steady-state concentration, there is an alternation of brain states expressed by EEG oscillations that vary between delta, theta, alpha, and burst suppression [6,7].

Imaging studies have shown remarkable similarities between the final state of unconsciousness in the anesthetized brain and the brain during deep NREM sleep [2]. It is known that synaptic and extrasynaptic gamma-aminobutyric acid (GABA) and N-methyl-D-aspartate (NMDA) receptors are the main targets of anesthetics, that the disruption of thalamo-cortical connectivity is a common feature of anesthetic action, and that thalamic deactivation from wakefulness to deep sleep is highly correlated with spindles and delta waves in the EEG [2]. Moreover, the cortex is profoundly deactivated (inhibited) during deep sleep and anesthetic-induced unconsciousness, and changes in cortical EEG are greater and more abrupt than those detected with a subthalamic electrode during the induction of anesthesia with propofol or sevoflurane [8]. The anesthetics act directly on cortical or thalamic neurons, inhibiting excitatory arousal pathways or potentiating the sleep pathways that control them. The initiation and maintenance of sleep is an active process that inhibits ascending arousal nuclei primarily through GABAergic inhibition from the hypothalamus and basal forebrain, and arousal nuclei can also send reciprocal inhibitory feedback to sleep-promoting nuclei [9,10].

Experimental evidence that sleep deprivation increases the efficacy of anesthetics [11], whereas prolonged anesthesia did not result in sleep deprivation [12], and that both sleep and anesthesia promote hypothermia [13], supports the notion that sleep and anesthesia share some common mechanisms [2].

Further evidence supports an interaction between endogenous sleep drive and anesthesia, showing that increased sleep drive increases sensitivity to anesthetics [14]. In addition, there is evidence that the sedative component of anesthesia is mediated by GABA**_A_** receptors in an endogenous sleep pathway [1]. GABAergic neurons of the ventrolateral preoptic nucleus (VLPO) are known to play an important role in the onset of sleep, and they project to brain arousal nuclei such as the histaminergic tuberomammillary nucleus, the serotonergic dorsal raphe, and the noradrenergic locus coeruleus [15,16,17]. Certain anesthetics have been shown to stimulate GABAergic neurons of the VLPO to exert their hypnotic effects [18]. In addition, the median preoptic nucleus (MnPO) is another nucleus that contains a high concentration of sleep-inducing GABAergic neurons [3]. Furthermore, MnPO neurons are more active during sleep deprivation, whereas VLPO neurons are more active during sleep, highlighting their different roles in sleep homeostasis, with MnPO responding to homeostatic pressure, whereas VLPO plays a more consolidative role [19].

Anesthesia and sleep onset may differ in terms of the neurotransmitters and molecular targets involved, as well as the anesthetic used [5]. It has been suggested that the type of GABAergic inhibition observed in both sleep and anesthesia has subtle variation in terms of the preferred subunit of the GABA receptor as a target, as well as the differential distribution of the subunit throughout the central nervous system, which may determine the site of action of anesthetics and natural somnogens [5]. Evidence suggests that neurons controlling the sleep−wake states (VLPO/MnPO GABAergic or glutamatergic neurons) do not necessarily mediate general anesthesia [4]. Apart from showing changes in EEG morphology with increasing depth of anesthesia and sleep that are typical of both sleep and surgical anesthesia, the anesthetic state shows a reduction in cerebral metabolism, a complete absence of nociceptive responses, a loss of protective reflexes (cough and gag reflex), and reduced postural control compared with sleep [5]. In addition, sleep, although characterized by amnesia like anesthesia, is thought to play a key role in memory consolidation and cognitive development [20,21]. In contrast with sleep, where sleep recovery is rapid (within minutes), resumption of wakefulness after general anesthesia can take up to hours [22]. Moreover, increased GABA-mediated inhibition and decreased glutamate-mediated excitation in the cortex during anesthesia are so strong that even high-intensity stimulation is insufficient to achieve transition to wakefulness, and the iatrogenic side effects of emergence from anesthesia (nausea, vomiting, cardiorespiratory depression, and loss of immune function) contrast with the refreshing nature of wakefulness after natural sleep [23,24,25].

It is known that the thalamus is a relay station that relays sensory inputs to the cortex, that the interaction between the thalamus and cortex shapes the electrophysiological characteristics of sleep, that many anesthetics inhibit thalamo-cortical neurons directly or through interaction with reticulo-thalamic neurons, and that loss of thalamo-cortical connectivity is considered a key event underlying loss of consciousness during both anesthesia and sleep [14]. Moreover, the reticulo-thalamic nucleus (RT) acts as a key brain structure for the local control of NREM sleep [26].

Because the GABAergic mechanism is an important target for anesthetic action and sleep promotion, as well as the hippocampus and RT being important brain structures for generating sleep oscillations (theta rhythm, sleep spindles, and delta rhythm) and control of unconscious states, in this study, we followed the alterations of hippocampal and RT GABAergic parvalbumin (PV) expressing interneurons as possible underlying mechanisms of the different local cortical and hippocampal EEG microstructures during NREM sleep compared with anesthesia-induced unconsciousness by two anesthetics with different main mechanisms of action (ketamine/diazepam versus propofol).

## 2. Results

### 2.1. Topography of EEG Microstructure during Distinct States of Unconsciousness

All differences in the local EEG microstructure in the motor cortex versus hippocampus during NREM sleep and anesthesia-induced unconsciousness with different mechanisms of action are shown in Figure 1.

Our results show that during unconsciousness induced by both anesthetics, in contrast with NREM sleep, the hippocampal delta amplitude was higher than that of the motor cortex (z ≥ −3.38; *p* ≤ 0.04). On the other hand, lower hippocampal gamma amplitude relative to motor cortex was a common feature of all three states of unconsciousness (z ≥ −3.44; *p* ≤ 0.05). In contrast with NREM sleep and propofol anesthesia, where hippocampal theta amplitude was augmented (z ≥ −3.48; *p* ≤ 0.01), it was attenuated during ketamine anesthesia relative to the motor cortex (z = −3.48; *p* = 10^−3^). In contrast with NREM sleep and ketamine anesthesia, where there were no differences between hippocampal and motor cortical sigma and beta amplitudes (z ≥ −1.81; *p* ≥ 0.07), hippocampal sigma and beta amplitudes were attenuated during propofol anesthesia relative to the motor cortex (z = −2.96; *p* ≤ 10^−3^).

Both anesthesia-induced states of unconsciousness are generally manifested by changes in delta and gamma, but clearly by changes in the amplitudes of theta, sigma, and beta EEG frequency bands. Whereas ketamine/diazepam-induced unconsciousness was topographically distinct only in delta and theta amplitudes, propofol-induced unconsciousness was distinct in delta, sigma, and beta amplitudes compared with NREM sleep.

### 2.2. Topography of EEG Microstructure during NREM Sleep Compared with Anesthesia-Induced Unconsciousness

The topographically represented EEG microstructure differences between three states of unconsciousness (NREM sleep, ketamine/diazepam-induced unconsciousness, and propofol-induced unconsciousness) are shown in Figure 2.

As we did not find significant differences between EEG microstructures during NREM sleep of both groups of rats before the induction of anesthesia in motor cortex (z ≥ −1.32; *p* ≥ 0.20) and hippocampus (z ≥ −1.85; *p* ≥ 0.07), we pooled all NREM sleep data for each brain structure as common NREM sleep of the motor cortex or hippocampus to further compare it with their EEG microstructures during the two anesthesia-induced unconscious states.

All three states of unconsciousness differed in the motor cortical and hippocampal EEG microstructures (χ^2^ ≥ 18.95; *p* ≤ 10^−4^ for motor cortex; χ^2^ ≥ 9.46; *p* ≤ 0.01 for hippocampus). Our results show that during propofol-induced unconsciousness, the attenuated delta and augmented sigma and beta amplitudes (z ≥ −4.13; *p* ≤ 0.01), followed by unchanged theta EEG amplitudes (z ≥ −1.87; *p* ≥ 0.06), were the globally expressed difference (at motor-cortical and hippocampal levels) compared with NREM sleep, while the increased gamma amplitude (z = −2.35; *p* = 0.02) was the only difference at the motor-cortical level compared with NREM sleep.

On the other hand, during ketamine/diazepam-induced unconsciousness, attenuated theta (z ≥ −5.53; *p* ≤ 10^−4^), increased beta and gamma amplitudes (z ≥ −4.82; *p* ≤ 10^−4^), and unchanged delta and sigma amplitudes (z ≥ −1.50; *p* ≥ 0.12) were the globally expressed difference from NREM sleep.

Moreover, both anesthesia-induced states of unconsciousness were common and globally expressed as increased beta amplitude (z ≥ −4.13; *p* ≤ 10^−3^) and increased motor cortical gamma amplitude (z ≥ −4.20; *p* ≤ 0.02) compared with NREM sleep.

The individual examples of the total hippocampal spectrograms (the overall 0–50 Hz frequency range) for each state of unconsciousness (Figure 3A), demonstrating EEG microstructure differences already presented as PDEs for each experimental group of rats in Figure 2, are shown together with their spectrograms for each frequency band in Figure 3B.

In addition, although the theta amplitude was globally unchanged during propofol anesthesia at the surgical level compared with NREM sleep (Figure 2), there was a significant increase in theta synchronization between the motor cortex and hippocampus compared with NREM sleep and ketamine/diazepam anesthesia (Figure 4, red arrow; z ≥ −3.89, *p* = 10^−4^). Moreover, there were the significant decreases in beta and gamma synchronizations between the motor cortex and hippocampus during ketamine/diazepam anesthesia at the surgical level versus NREM sleep and propofol anesthesia (Figure 4; blue arrows; z ≥ −3.59, *p* ≤ 0.02).

### 2.3. Alteration of GABAergic PV-Expressing Interneurons in the Hippocampus and RT during Different States of Unconsciousness

In this study, we compared the differences in local EEG microstructures during simultaneous NREM sleep of the hippocampus and motor cortex compared with two states of unconsciousness induced by anesthetics with different mechanisms of action, and we specifically tracked GABAergic PV-expressing interneurons (PV+) in the hippocampus and RT.

Our results show a significant suppression of PV expression in the dentate gyrus of the hippocampus (DG) (z = −2.71; *p* = 0.01) and RT in all rats during ketamine/diazepam anesthesia compared with propofol anesthesia. Figure 5A shows the average number of PV+ interneurons in DG of the ketamine/diazepam-anesthetized rat group (n = 9) compared with the propofol-anesthetized rat group (n = 10). Figure 5B shows three typical single examples of PV+ interneurons in DG for each experimental group. 

In addition to this ketamine/diazepam-induced suppression of PV+ interneurons in DG, there was also suppression of PV+ interneurons in the CA3 region of the hippocampus (z = −4.16; *p* = 10^−4^; Figure 6) and in the RT (Figure 7) in the ketamine/diazepam (n = 9) versus propofol group (n = 10) of rats. The suppression of PV+ interneurons in the RT even showed up in the form of defects in PV immunostaining in the ketamine/diazepam group compared with the propofol group (Figure 7, top two rows).

### 2.4. Changes in PSD-95 Expression during Different PV Expression in the Hippocampus and RT in Different States of Unconsciousness

In this study, we used PSD-95 as an excitatory synaptic marker to test for a change in local excitation following changes in PV expression in the hippocampus or RT.

Whereas PSD-95 expression increased in DG (Figure 8; especially in the suprapyramidal granule cell layer of DG), there was no difference in PSD-95 expression in RT (Figure 9) during ketamine/diazepam anesthesia compared with propofol anesthesia.

Our results show that the suppression of PV expression is followed by an inhibitory/excitatory imbalance only in the hippocampus during ketamine/diazepam unconsciousness (Figure 8 and Figure 9).

## 3. Discussion

Our study shows the differential expression of PV at the level of the hippocampus and RT during ketamine/diazepam and propofol-induced unconsciousness. Therefore, the suppression of PV expression during ketamine/diazepam anesthesia, in addition to the different main mechanisms of action, might be an additional mechanism underlying the different local and global EEG microstructures during this unconsciousness compared with propofol-induced unconsciousness and NREM sleep, in particular expressed by the global theta attenuation and hippocampal gamma augmentation.

Whereas both states of anesthesia-induced unconsciousness compared with NREM sleep were expressed by globally increased beta and cortically increased gamma amplitude, globally attenuated theta and increased hippocampal gamma amplitude were the hallmarks of ketamine/diazepam-induced unconsciousness compared with propofol-induced unconsciousness and with NREM sleep, respectively.

On the other hand, the hallmark of propofol-induced unconsciousness was globally attenuated delta and augmented sigma amplitude compared with ketamine/diazepam-induced unconsciousness and NREM sleep.

There is evidence that the cortical [27] and brainstem [28] effects of ketamine are different from those of isoflurane and propofol, which enhance GABA**_A_**ergic inhibition. The main mechanism of ketamine’s anesthetic effects is mediated by NMDA receptors [29], and the blockade of NMDA receptors appears to preferentially affect GABAergic over glutamatergic cells, resulting in excitatory effects at cortical and other brain areas [30]. Moreover, unlike GABA**_A_**ergic anesthetics, ketamine stimulates cholinergic, monoaminergic, and orexinergic arousal systems [28]. In addition, ketamine is a hypnotic and analgesic that stimulates respiration and abolishes the coupling between unconsciousness and upper airway muscle dilation [31], whereas GABAergic anesthetics strongly depress respiration. Because we used a mixture of ketamine and diazepam in our study, the global beta and cortical gamma augmentation that is a common feature of both anesthetic-induced states of consciousness compared with NREM sleep could be a consequence of a common GABA_A_ agonist effect. Nevertheless, the globally attenuated theta and increased hippocampal gamma amplitude that is the hallmark of ketamine/diazepam-induced unconsciousness was the single most important NMDA-antagonistic effect of ketamine. Whereas NMDA blockade was manifested at the cortical and hippocampal levels as the attenuated theta amplitude, at the hippocampal level, attenuated theta amplitude was followed by an increased gamma amplitude.

Our results suggest that this NMDA blockade was preferentially mediated on GABAergic interneurons in the hippocampus and RT during ketamine/diazepam- versus propofol- induced unconsciousness during stable surgical anesthesia.

In Sprague-Dawley rats, there is evidence that ketamine increases acetylcholine release in the hippocampus and frontal cortex, in addition to its main mechanism of action, with no effect on the striatum [32]. This important role of ketamine in acetylcholine release in the hippocampus is explained by the demonstration of the highest density of NMDA receptors in the rat hippocampus [33]. Ketamine is a unique anesthetic that interacts with NMDA, opioid, monoaminergic, and muscarinic receptors and voltage-sensitive Ca**^++^** channels, but not with GABA receptors [34]. In addition, there is evidence that the activity of locus coeruleus noradrenergic neurons facilitates ketamine anesthesia and that the noradrenergic content of the cortex and hippocampus is correlated with the duration of ketamine anesthesia compared with thiopental anesthesia [14]. Furthermore, global neuronal inhibition caused by different mechanisms of action of the anesthetic results in a marked complexity gradient between brain structures, with EEG complexity (neuronal activity) at the cortical and hippocampal levels being higher for ketamine/xylazine anesthesia than for pentobarbital anesthesia, with the hippocampal level dominating in rats [35]. In contrast with isoflurane, sevoflurane, propofol, and pentobarbital, which decreased the activity (c-fos expression) of hypocretin/orexin neurons, this was not the case with ketamine [14].

Furthermore, our results show that hippocampal PV suppression is followed by locally increased excitation (increased PSD-95 expression) during unconsciousness induced by ketamine/diazepam, and that in addition to global theta amplitude attenuation and hippocampal gamma amplitude augmentation, there may also be a decrease in cortical and hippocampal synchrony in the theta, beta, and gamma frequency ranges compared with NREM sleep and propofol anesthesia-induced unconsciousness. On the other hand, global delta amplitude attenuation and sigma augmentation along with increased cortico-hippocampal theta synchronization are the unique features of propofol-induced unconsciousness compared with NREM- and ketamine/diazepam-induced unconsciousness. This finding is consistent with other in vivo evidence and mathematical modeling for the role of thalamo-cortical synchronization as a mechanism for propofol-induced unconsciousness through increasing the potentiation of GABAergic inhibition [36,37]. It is known that RT is the only inhibitory part of the cortico-thalamo-cortical network that plays an important role in sleep homeostasis, cortical rhythm generation, attention, and consciousness [36]. Although burst-firing of RT neurons is involved in the generation of cortical sleep spindles through rhythmic inhibition of excitatory thalamo-cortical neurons, it is also involved in the generation of the delta oscillation [38,39], but there are few data to date related to dysfunction of the inhibitory neurons of RT and altered theta rhythm, except for some indirect data related to its role in attention [40], knockout of RT-selectively expressing genes and enhanced theta rhythm in attention deficit hyperactivity disorder (ADHD) in mice, and enhanced theta rhythm in ADHD patients [38,41,42]. In contrast with the hippocampus, our study shows PV suppression in the RT only during unconsciousness induced by ketamine/diazepam, but this PV suppression was not followed by locally increased excitation (increased PSD-95 expression in the RT).

We show that PV suppression in the hippocampus and RT underlies global theta amplitude attenuation and hippocampal gamma augmentation along with decreased cortico-hippocampal theta, beta, and gamma synchronization, which is a unique feature of ketamine-induced unconsciousness compared with propofol-induced unconsciousness and NREM sleep.

Because the GABAergic mechanism is an important target for the action of anesthetics and the promotion of sleep, and the hippocampus and RT are important brain structures for the generation of sleep oscillations and the control of unconsciousness, we followed the underlying GABA PV interneurons. When we detected a distinct PV+ change that did not reveal anything about neuronal activity, but only a local functional change in the GABA PV interneurons, we additionally tested the local inhibitory/excitatory balance using PSD-95 as an excitatory synaptic marker. Furthermore, everything we did was performed at the surgical level using different anesthesia. The level of anesthesia was tracked at the behavioral level, and both groups of rats were behaviorally under the same stable surgical anesthetic level.

Moreover, although PV suppression led to an increase in hippocampal PSD-95 expression, it did not lead to an imbalance between inhibition and excitation at RT during ketamine/diazepam anesthesia compared with propofol anesthesia. This increased excitation in the hippocampus could be a consequence of the lower GABA interneuronal activity caused by the synergistic GABA**_A_** agonistic and NMDA antagonistic inhibitory effects of ketamine/diazepam anesthesia and an additional mechanism underlying the unique local EEG microstructure in the hippocampus during ketamine/diazepam anesthesia: the local attenuation of theta amplitude and increase in gamma amplitude in the hippocampus during ketamine/diazepam anesthesia compared with propofol anesthesia. Our results show that PV suppression in RT during ketamine/diazepam anesthesia does not alter the local level of excitation in RT, which was already equally abolished by both anesthetics. Furthermore, the same level of anesthesia in this study was evidenced at the molecular level by the same PSD-95 expression in RT (the same local level of arousal evidenced by excitatory biomarker), although the suppression of PV+ in RT occurred during ketamine/diazepam anesthesia versus propofol anesthesia.

Our study shows that although anesthesia and sleep share many neurobiological features, they are distinct states in terms of local EEG microstructure and its underlying GABAergic and molecular substrate in different brain structures (distinct local neuronal networks) that are important for unconsciousness and EEG rhythm formation.

## 4. Materials and Methods

### 4.1. Experimental Design

We used 20 adult male Wistar rats (2.5 months old, weighing between 250 and 290 g) chronically implanted for sleep recording. After 3 h of sleep recording, the rats were divided into two experimental groups: one half of the rats (n = 10) received ketamine/diazepam anesthesia (100 mg/kg, Zoletil^®^ 50, VIRBAC, Carros, France) and the other half of the rats (n = 10) received propofol anesthesia (100 mg/kg; Propofol Lipuro 2% (20 mg/mL), B. BRAUN ADRIA RSRB d.o.o., Belgrade, Serbia) intraperitoneally. After achieving surgical anesthesia, we simultaneously recorded the EEG of the motor cortex and hippocampus for 1 h, and finally, each rat was transcardially perfused and sacrificed for further immunohistochemical procedures.

After surgery and throughout the experimental protocol, animals were housed individually in custom-made clear plexiglas cages (30 × 30 × 30 cm) on a 12-h light−dark cycle (lights on at 07:00 a.m., lights off at 19:00 p.m.) at 25 °C and received food and water ad libitum.

### 4.2. Surgical Procedure

Surgical procedures for chronic electrode implantation for sleep recording were performed as previously described [43,44,45,46].

Briefly, the rats were anesthetized with ketamine/diazepam (50 mg/kg, i.p., Zoletil^®^ 50, VIRBAC, Carros, France) and positioned in a stereotaxic frame (Stoelting Co., Dublin, Ireland). For EEG recordings, we implanted two stainless-steel epidural screw electrodes in the motor cortex (MCx; A/P: + 1.0 mm from bregma; R/L: 2.0 mm from sagittal suture; D/V: 1.0 mm from the skull) and two wire electrodes (teflon-coated stainless steel wire, Medwire, NY, USA) in the CA1 hippocampal regions (Hipp; A/P: −3.6 mm from bregma; R/L: 2.5 mm from sagittal suture; D/V: 2.5 mm from brain surface), in agreement with Paxinos and Watson [47]. To assess skeletal muscle activity (EMG), the bilateral wire electrodes were implanted in the dorsal neck muscles, and a stainless-steel screw electrode was implanted as a mass in the nasal bone. All electrode leads were soldered to a miniature connector (39F1401, Newark Electronics, Schaumburg, IL, USA), and the assembly was attached to the screw electrodes and to the skull with acrylic cement (Biocryl-RN, Galenika a.d. Beograd, Serbia).

### 4.3. Recording Procedure

All sleep recordings were performed 14 days after the surgical procedure. Sleep was recorded for 3 h during the light phase, starting at 09:00 a.m. EEG and EMG activities were recorded differentially. After conventional amplification and filtering (0.3–100 Hz bandpass; A-M System Inc. model 3600, Carlsborg, WA, USA), the analog data were digitized (at a sampling rate of 256/s) using DataWave SciWorks Experimenter version 11.2 (A-M System, Carlsborg, WA, USA), and the EEG and EMG activities were displayed on a computer monitor and stored on a hard disk for further offline analysis [43,44,45,46]. After 3 h of sleep recording, the rats were anesthetized with ketamine/diazepam (100 mg/kg, Zoletil^®^ 50, VIRBAC, France) or propofol (100 mg/kg; Propofol Lipuro 2% (20 mg/mL), B. BRAUN ADRIA RSRB d.o.o., Belgrade, Serbia) and we simultaneously recorded EEGs of the motor cortex and hippocampus during 1 h of stable surgical stage of both anesthetics. Stability of anesthesia was estimated from the observed loss of consciousness; muscle atonia; absence of tail pinch, ear pinch (analgesia), and corneal reflexes; and respiratory pattern before each recording of the stable anesthetic state of the surgical level.

### 4.4. Local EEG Analysis during NREM Sleep and Anesthesia-Induced Unconsciousness

All of the EEG analyses were performed in MATLAB R2011a (MathWorks Inc., Natick, MA, USA) using software originally developed in MATLAB 6.5 [43,44,45,46,48]. The EEG analysis of the individual states of unconsciousness included only the artifact-free recorded signals. In this study, we applied the FFT algorithm to the signals acquired during each 3 h sleep recording (a total of 1080 10 s Fourier epochs) and automatically differentiated each 10 s epoch as a waking state, NREM state, or REM state [44,45,46,48]. Specifically, to assess local NREM sleep, we extracted 30 min of simultaneous NREM sleep (a total of 180 simultaneous NREM 10 s epochs from motor cortex and hippocampus), obtained from previously defined total simultaneous NREM sleep of the motor cortex and hippocampus, extracted always from the first and the second hour of all sleep recordings [43,44,46]. Then, we analyzed the local NREM sleep EEG microstructure by calculating the relative NREM amplitudes of all conventional EEG frequency bands (δ = 0.3–4 Hz; θ = 4.1–8 Hz; σ = 10.1–15 Hz; β = 15.1–30 Hz; γ = 30.1–50 Hz) [43,44,45]. In addition, the local NREM sleep EEG microstructures of the motor cortex and hippocampus were compared with their EEG microstructures during 30 min of unconsciousness induced by a specific anesthetic. To analyze the EEG amplitude differences during NREM sleep (physiological unconsciousness) and each anesthesia (different states of unconsciousness induced by anesthetics with different mechanisms of action), we calculated the group probability density distributions of the relative amplitudes over 30 min of NREM sleep or distinct anesthesia using the probability density estimation (PDE) routine provided by MATLAB R2011a [43,44,45,46,49]. PDE analysis was performed on the assembles of the relative amplitudes by pooling the measured values from all of the animals belonging to the specific experimental group. Finally, group mean values of each frequency band relative amplitude were calculated for each 15 min.

We also calculated the cortico-hippocampal coherences by using the “mscohere” routine of the MATLAB R2011a Signal Processing Toolbox, which computes the magnitude of squared coherence between the signals. For this purpose, the individual 30 min EEG signals, derived from the motor cortex and hippocampus, were concatenated and pooled within each unconsciousness state (NREM sleep, and ketamine/diazepam- or propofol-induced unconsciousness). Coherence values between 0 and 1 were calculated for every 10 s of FFT epoch and each frequency point (at a 0.1 Hz frequency resolution), within the overall 0.3–50 Hz range. Then, the values within each conventional frequency band were averaged. Finally, the group mean coherence values were calculated for each 15 min, each frequency band, and each experimental group [48].

### 4.5. Tissue Processing for PV and PSD-95 Immunohistochemistry

At the end of each recording of stable anesthesia at the surgical level, the rats were sacrificed for further PV and PSD-95 immunohistochemistry. All of the animals were perfused transcardially with 0.9% saline, then with 4% paraformaldehyde (PFA, Sigma-Aldrich, Taufkirchen, Germany) in 0.1 M phosphate-buffered saline (PBS, pH = 7.4), and finally with a 10% sucrose solution in 0.1 M PBS. The brains were removed and placed in 4% PFA overnight and then in 30% sucrose solution for several days. The brains were serially sectioned into coronal 40 μm thick sections in a cryostat (Thermo Fisher NX70, MA, USA), and the free-floating sections were stored in a cryoprotective buffer for further use [46,50].

The free-floating brain sections were first thoroughly rinsed with 0.1 M PBS (pH 7.4). Non-specific binding was prevented by incubation in 3% hydrogen peroxide/10% methanol/0.1 M PBS for 15 min and 5% normal donkey serum/0.1 M PBS (D9663, Sigma-Aldrich, Burlington, MA, USA) for 60 min at room temperature. The sections were then incubated overnight at +4 °C with a mouse monoclonal anti-PV antibody (dilution 1:2000, P3088, Sigma-Aldrich, Burlington, MA, USA) and mouse monoclonal anti-PSD-95 antibody (dilution 1:200, MAB1598, Merck Millipore, Burlington, MA, USA). The primary antibodies were diluted in PBS with 0.3% Triton X-100. After three 5 min washes in 0.1 M PBS, the sections were incubated with polyclonal rabbit anti-mouse immunoglobulin (dilution 1:100, Agilent Dako, P0260, Glostrup, Denmark) for 90 min. Immunoreactive signals were visualized with a diaminobenzidine solution (1% 3,3-diaminobenzidine [11208, Acros organics, Geel, Belgium]/0.3% hydrogen peroxide/0.1 M PBS). All of the sections were finally mounted on slides, air dried, dehydrated with increasing alcohol concentrations (70%, 96%, 100% ethanol, Zorka Pharma, Sabac, Serbia), placed in a clearing agent (Xylene, Zorka Pharma, Sabac, Serbia), coverslipped with DPX (Sigma-Aldrich, Burlington, MA, USA), and examined under a Leica light microscope with camera (Leica DMRB, Wetzlar, Germany). To test the specificity of the immunostaining, the primary antibodies were omitted from the control experiments.

Quantification of PV immunoreactivity was performed in the DG and CA3 using ImageJ 1.46 software (NIH, Bethesda, MD, USA) and by manually counting the number of PV+ interneurons. The number of PV+ interneurons was counted in DG and CA3 per brain side of each rat within the stereotaxic range of −3.10 to −3.50 mm posterior to bregma [51], drawn for each experimental group, and expressed as mean number + SE.

### 4.6. Statistical Analysis

All of the statistical analyzes were performed using a Kruskal−Wallis ANOVA (χ**^2^** values) with a Mann−Whitney U (z values) two-sided post hoc test. The accepted significance level was *p* ≤ 0.05 in all of the cases.

## Figures and Tables

**Figure 1 ijms-24-06769-f001:**
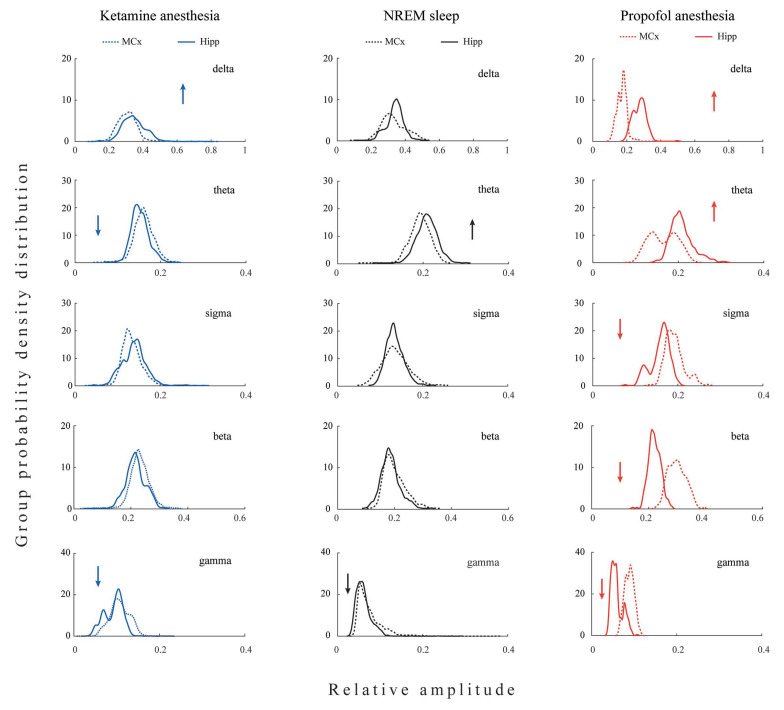
Topography of EEG microstructure during distinct states of unconsciousness. The group probability density distributions (PDE)/30 min of the relative amplitudes of all conventional EEG frequency bands (delta, theta, sigma, beta, and gamma) in the motor cortex (MCx) compared with the hippocampus (Hipp). PDE data for each brain structure and each state of unconsciousness are pooled from all rats belonging to the corresponding experimental group (NREM sleep: MCx n = 12 and Hipp n = 13; ketamine/diazepam anesthesia: MCx n = 6 and Hipp n = 7; propofol anesthesia: MCx: n = 6 and Hipp n = 7). Arrows show the statistically significantly higher (up arrows) or lower (down arrows) amplitude of a particular EEG frequency band in the hippocampus compared with the motor cortex at *p* ≤ 0.05.

**Figure 2 ijms-24-06769-f002:**
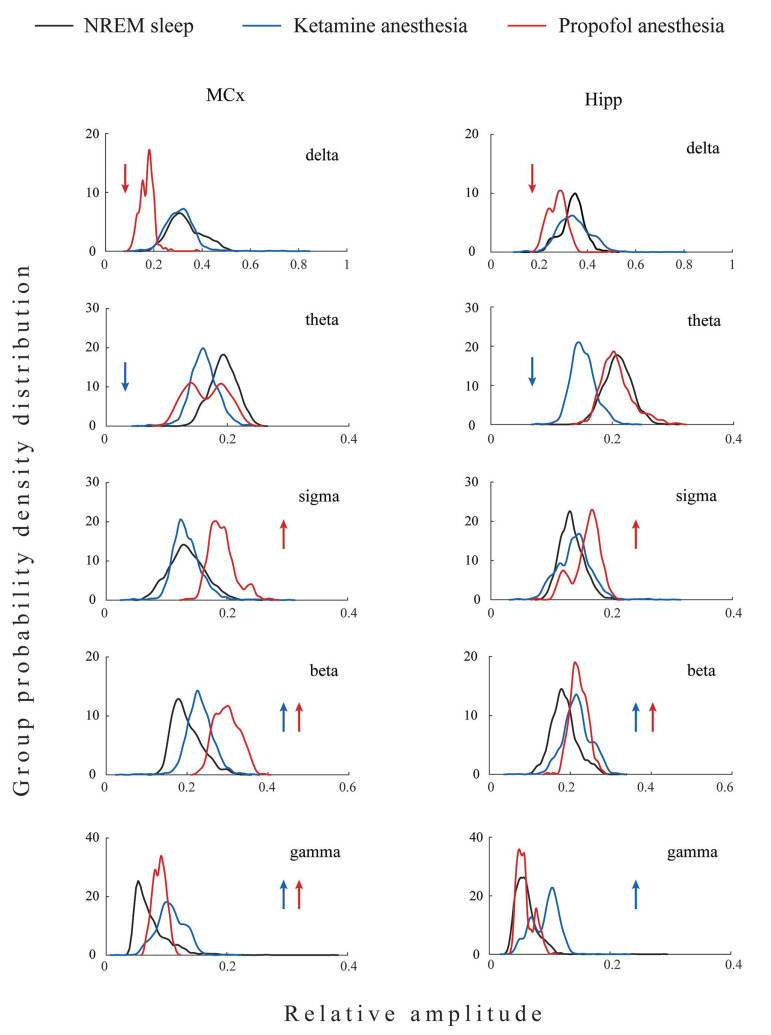
Topography of the EEG microstructure during NREM sleep compared with anesthesia-induced unconsciousness. The group probability density distributions (PDE)/30 min of the relative amplitudes of all conventional EEG frequency bands (delta, theta, sigma, beta, and gamma) of the motor cortex (MCx) and hippocampus (Hipp) during NREM sleep compared with ketamine/diazepam and propofol anesthesia-induced unconsciousness. PDE data for each brain structure and each state of unconsciousness are pooled from all rats belonging to the corresponding experimental group (NREM sleep: MCx n = 12 and Hipp n = 13; ketamine/diazepam anesthesia: MCx n = 6 and Hipp n = 7; propofol anesthesia: MCx n = 6 and Hipp n = 6). Arrows show a statistically significant increase (up arrows) or decrease (down arrows) in the amplitude of a given EEG frequency band in the corresponding experimental group at *p* ≤ 0.05.

**Figure 3 ijms-24-06769-f003:**
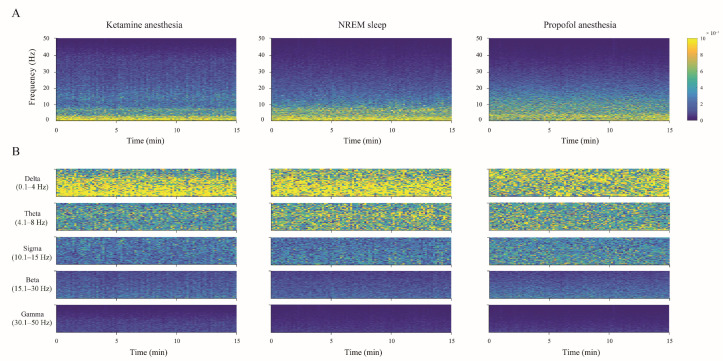
Hippocampal spectrograms during different states of unconsciousness. The individual examples of the total hippocampal spectrograms (the overall 0–50 Hz frequency range) during 15 min of each state of unconsciousness (NREM sleep, ketamine/diazepam, and propofol anesthesia) (**A**) with their spectrograms for each frequency band (**B**). The color bar is the same for all spectrograms.

**Figure 4 ijms-24-06769-f004:**
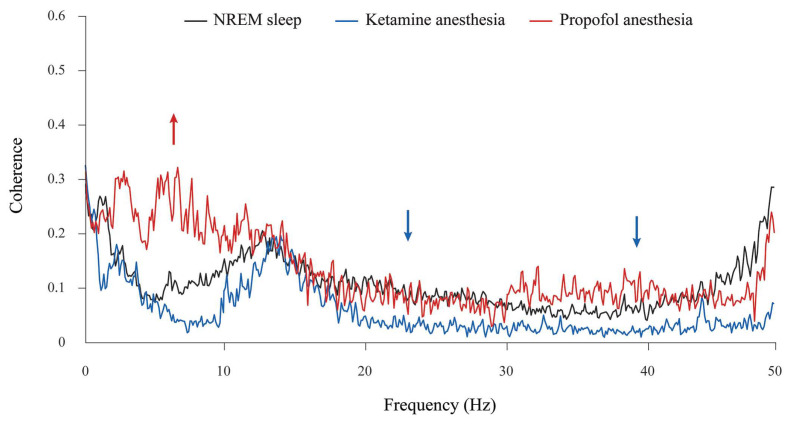
Cortico-hippocampal coherence spectra during different states of unconsciousness. Mean coherence spectra during NREM sleep (n = 12), ketamine/diazepam anesthesia (n = 6), and propofol anesthesia (n = 6). Arrows show a statistically significant increase (up arrows) or decrease (down arrows) in cortico-hippocampal coherence within the specific EEG frequency band in the corresponding experimental group at *p* ≤ 0.05. While a significant increase in theta synchronization (red arrow) is observed only during propofol anesthesia compared with NREM sleep and ketamine/diazepam anesthesia, there are significant decreases in beta and gamma synchronizations (blue arrows) during ketamine/diazepam anesthesia compared with NREM and propofol anesthesia.

**Figure 5 ijms-24-06769-f005:**
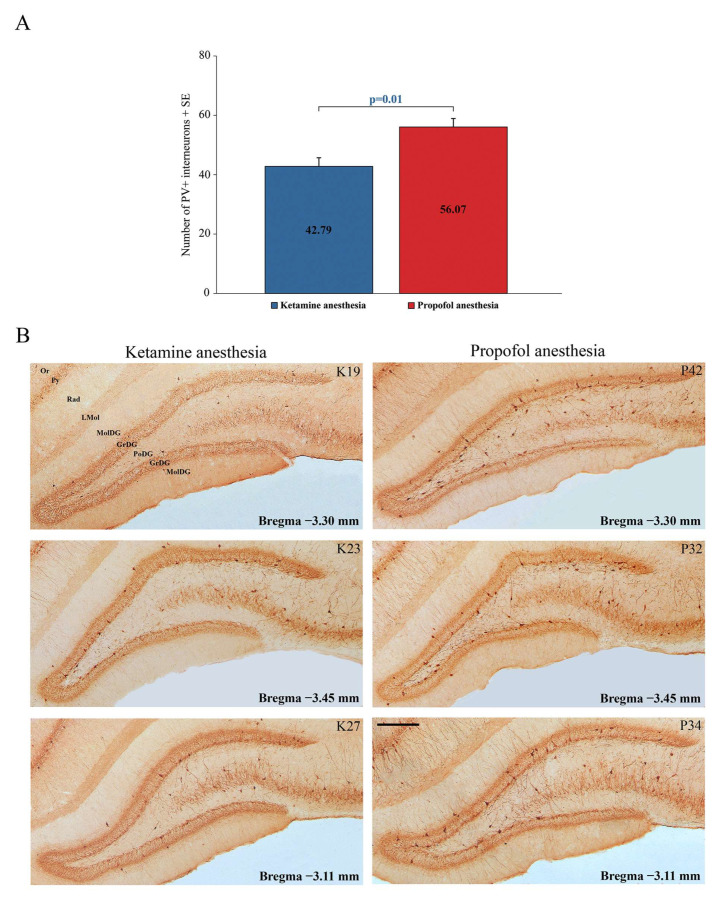
Suppression of PV+ interneurons in the hippocampal DG. (**A**) Mean number of PV+ interneurons in the hippocampal DG of the ketamine/diazepam-anesthetized rat group (n = 9) versus the propofol-anesthetized rat group (n = 10). (**B**) Typical individual examples of PV suppression within the hippocampal DG of the rats anesthetized with ketamine/diazepam (K19, K23, and K27) versus those anesthetized with propofol (P32, P34, and P42). Or—oriens layer of the hippocampus; Py—pyramidal cell layer of the hippocampus; Rad—radiatum layer of the hippocampus; LMol—lacunosum moleculare layer of the hippocampus; MolDG—molecular layer of the dentate gyrus; GrDG— granule cell layer of the dentate gyrus; PoDG— polymorph cell layer of the dentate gyrus. Scale bar is 200 µm.

**Figure 6 ijms-24-06769-f006:**
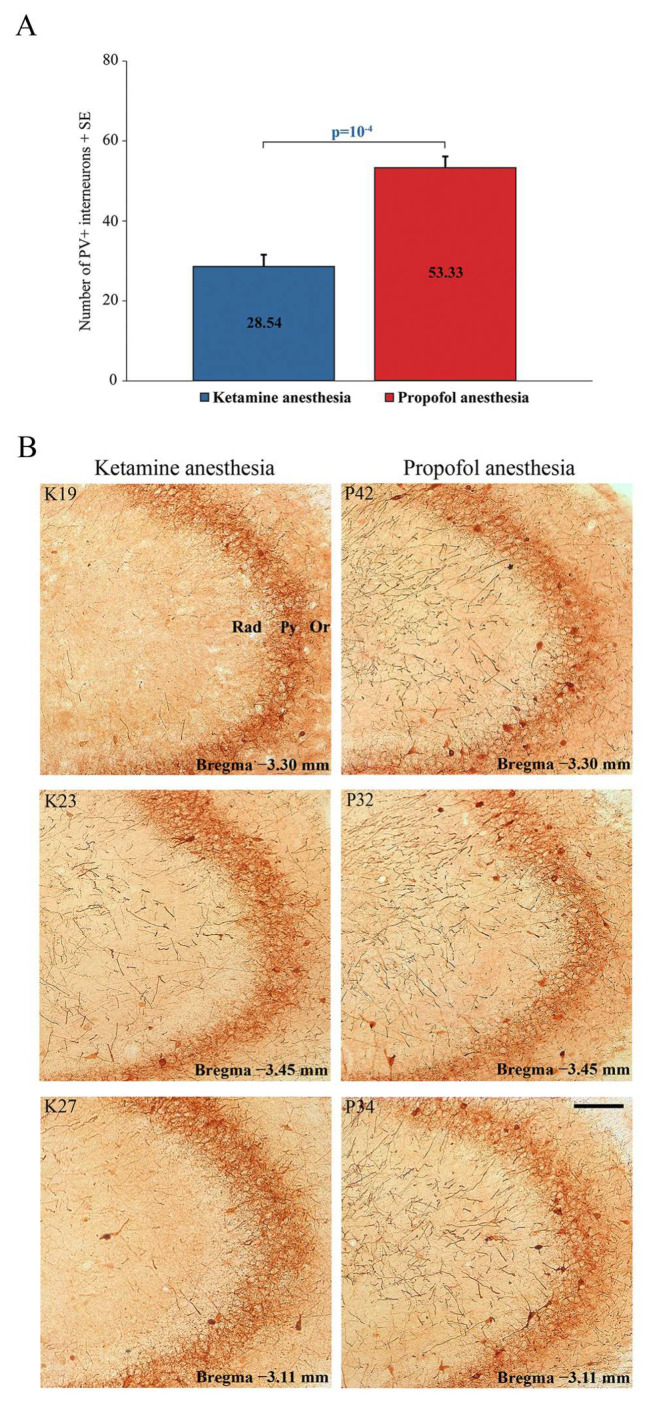
Suppression of PV+ interneurons in the hippocampal CA3 region. (**A**) Mean number of PV+ interneurons in the hippocampal CA3 of the ketamine/diazepam-anesthetized rat group (n = 9) versus the propofol-anesthetized rat group (n = 10). (**B**) Typical individual examples of PV suppression within CA3 region of the hippocampus in the ketamine/diazepam (K19, K23, and K27) versus propofol group of rats (P32, P34, and P42) from Figure 5. Or—oriens layer of the hippocampus; Py—pyramidal cell layer of the hippocampus; Rad—radiatum layer of the hippocampus. Scale bar is 100 µm.

**Figure 7 ijms-24-06769-f007:**
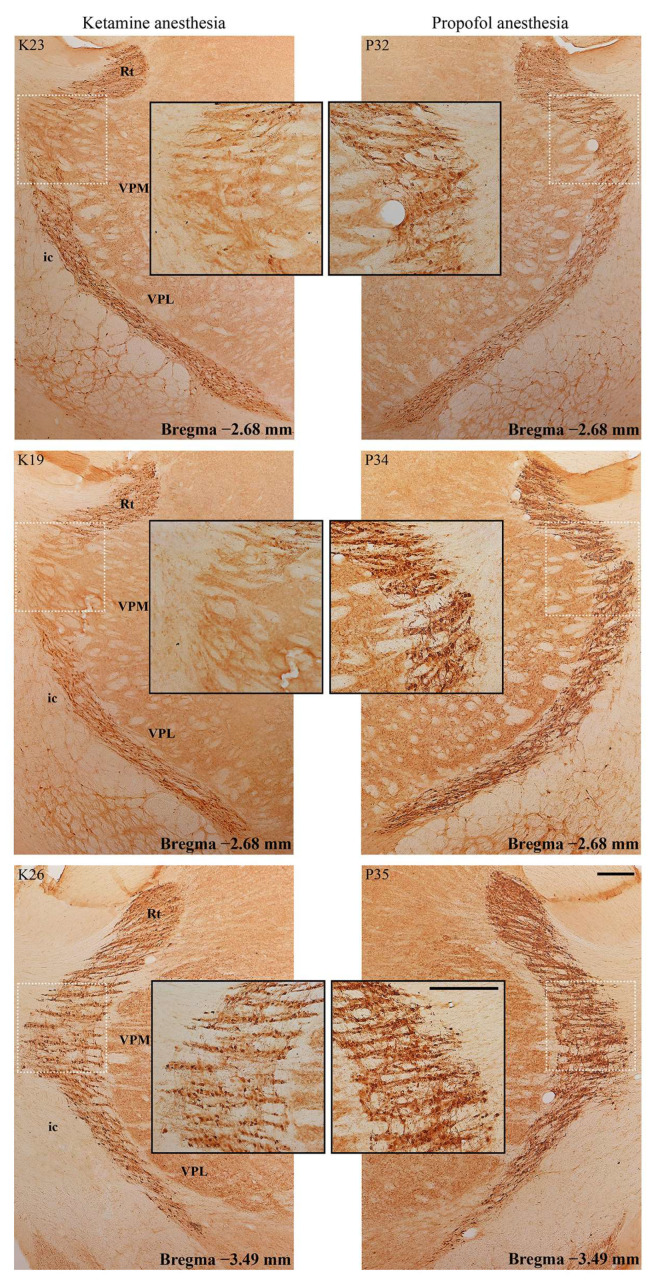
Suppression of PV+ interneurons in the reticulo-thalamic nucleus (RT). Typical individual examples of PV suppression within RT in the ketamine/diazepam (K19, K23, and K26) versus propofol group of rats (P32, P34, and P35). Rt—reticular nucleus; VPM—ventral posteromedial thalamic nucleus; VPL—ventral posterolateral thalamic nucleus; ic—internal capsule. The scale bar is 200 µm and for the inserts it is 100 µm.

**Figure 8 ijms-24-06769-f008:**
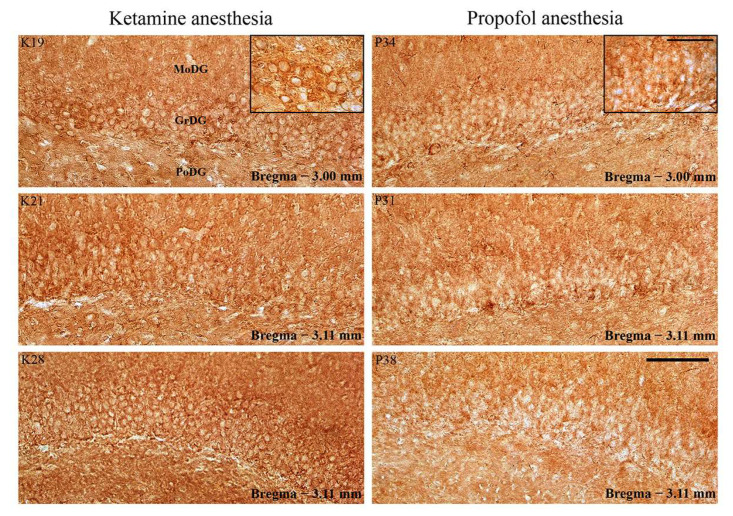
PSD-95 expression in the DG of the hippocampus. Individual examples of the increased PSD-95 expression in the suprapyramidal granule cell layer of DG during PV suppression in the ketamine/diazepam anesthesia (K19, K21, and K28) versus propofol anesthesia (P31, P34, and P38). MolDG—molecular layer of the dentate gyrus; GrDG—granule cell layer of the dentate gyrus; PoDG—polymorph cell layer of the dentate gyrus. Scale bar is 50 µm and for the inserts it is 16 µm.

**Figure 9 ijms-24-06769-f009:**
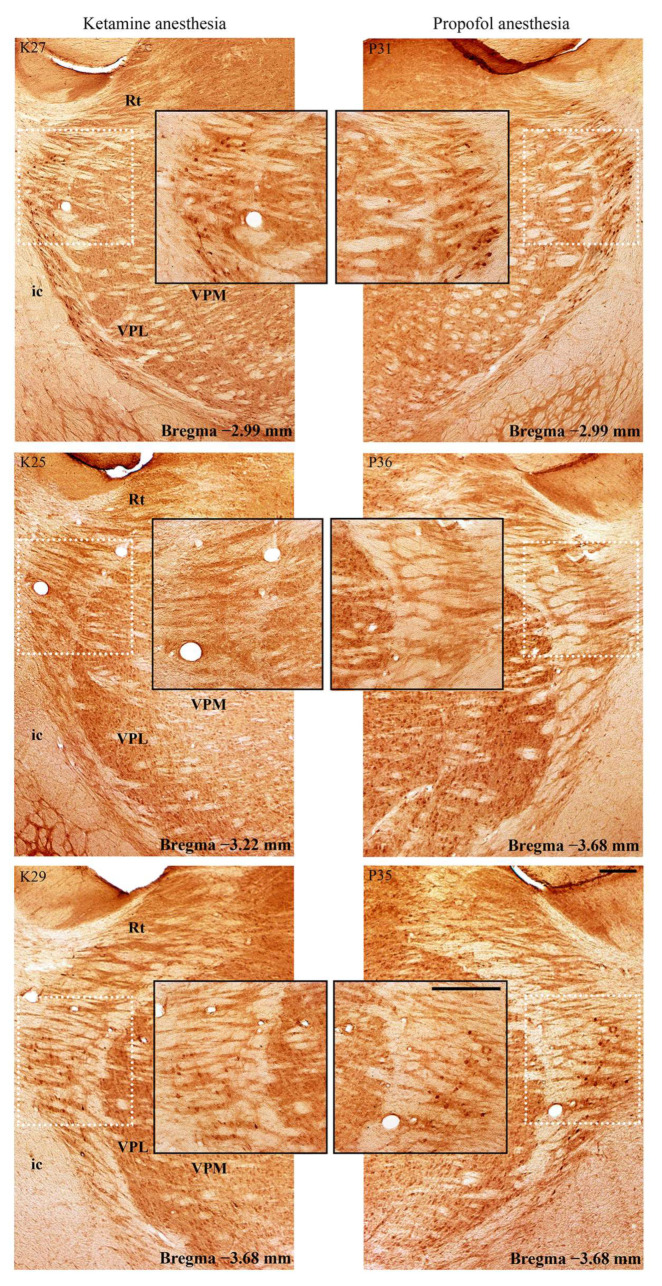
PSD-95 expression in RT. Individual examples PSD-95 expression in the RT during ketamine/diazepam anesthesia (K25, K27, and K29) versus propofol anesthesia (P31, P35, and P36). There is no difference in PSD-95 expression. Rt—reticular nucleus; VPM—ventral posteromedial thalamic nucleus; VPL—ventral posterolateral thalamic nucleus; ic—internal capsule. The scale bar is 200 µm and for the inserts 100 µm.

## Data Availability

Not applicable.

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
