# Peer review of "Different Alterations of Hippocampal and Reticulo-Thalamic GABAergic Parvalbumin-Expressing Interneurons Underlie Different States of Unconsciousness"

_ijms, 2023, doi:10.3390/ijms24076769_

Round 1
Reviewer 1 Report
Reviewer Comment
This study compared the expression of parvalbumin (PV) -expressing interneurons in the hippocampus and at the reticulo-thalamic nucleus (RT) under different states of unconsciousness and found that ketamine/diazepam anesthesia does not abolish the local level of excitation in RT. By investigating the structural changes and EEG patterns with different types of unconsciousness, the authors tried to understand the differences in mechanisms between anesthesia induced by different anesthetics and sleep. Although the result is interesting, many issues should be addressed to clarify the main findings.
Major comment,
1. Firstly, why did the authors use a mixture of ketamine/diazepam instead of ketamine alone? It is important to consider the potential impact of diazepam use on the EEG and immunohistochemistry results. How do you think the use of diazepam may have affected the outcome of this study?
2. The authors compared the EEG results in the motor cortex and hippocampus. In this study, were the levels of anesthesia with ketamine/diazepam equivalent to those achieved with propofol anesthesia? If there were differences in anesthetic depth, these differences should be considered when interpreting the EEG and immunohistochemistry results. How was the dosage of anesthetics (diazepam/ketamine vs. propofol) selected?
3. According to the study protocol, after waking up from sleep, the rat was anesthetized with one of the two selected anesthetic protocols using randomization. Before anesthesia, was the rat fully awake from sleep? How can you ensure that the rats did not have any residual effect from sleep?
4. Page 15, line 410 “.. included only artifact-free recorded signals…”. How were artifact-free signals selected in this study? Were there any preprocessing steps before analysis?
5. The authors observed PV suppression in the hippocampus and at RT, as well as hippocampal PSD-95 expression increase, which was not observed under ketamine anesthesia. From these results, it appears that the structural changes are likely to be related to EEG changes under different states of unconsciousness. However, several questions arise from these results. Is the overexpression a result of the recruitment of additional neurons or functional activation? Does the alteration of expression return to normal after recovery from anesthesia? From the perspective of the unconsciousness mechanism, how did you interpret the alteration of expression pattern, and how can we interpret it to understand the EEG results?
6. In the discussion section, the majority of the discussion focuses on reviewing previous literature. While it is important to consider the results in the context of the literature, more space should be devoted to discussing the results. For example, why does the alteration of interneuron expression/activity underlie different states of unconsciousness, even though anesthesia and sleep induce unconsciousness, which is indistinguishable from each other?
Minor comment
1. In Figures 6, 7, 8, and 9, the authors should add the results quantifying the number of positive cells, similar to Figure 5A. For example, it is difficult to see PV positive suppression in the hippocampal CA 3 region in Figure 6.
2. The x-axis of Figures 1 and 2 is unclear. It would be helpful to know what the x-axis represents and whether the numbers on the x-axis are relative number.
3. The manuscript should describe the range of each frequency band (delta, theta, alpha, beta, gamma) used in the study.
4. In Figure 3B, it is unclear whether the scale bar is the same as in Figure 3A. The authors should clarify this.
Reviewer 2 Report
At the manuscript "Different alterations of hippocampal and reticulo-thalamic GABAergic parvalbumin-expressing interneurons underlie different states of unconsciousness,” by Drs. Ljiljana Radovanovic et al authors describe the results of the study of the changes in hippocampal and reticulothalamic nucleus (RT) neurons expressing GABAergic parvalbumin (PV). It was assumed that the mechanisms involved underlie various local cortical and hippocampal electroencephalographic (EEG) changes during sleep with rapid eye movements (NREM). The authors have done a very impressive study, the findings are important for our understanding of the mechanisms of consciousness and anesthesia.
I have no objection to the essence of the manuscript, there are only some questions.
The issues that the authors studied using experiments on animals were also studied in clinical practice. Usually "loss of consciousness" (LoC) and "recovery of consciousness" (RoC) in medicine is determined using behavioral correlates. But physiological correlates of LoC and RoC are also very important, these correlates have been investigated in experiments with propofol. Are there parallels here, according to the authors? I would advise you to use these publications (below). The point of view of the authors would be very interesting.
Subramanian et al, Detecting loss and regain of consciousness during propofol anesthesia using multimodal indices of autonomic state, Proc. Annu. Int. Conf. IEEE Eng. Med. Biol. Soc. EMBS (2020) 824–827, doi.org/10.1109/EMBC44109.2020.9175366.
Tsytsarev, Methodological aspects of studying the mechanisms of consciousness. Behav Brain Res. 2022 Feb 15;419:113684. doi: 10.1016/j.bbr.2021.113684.
Flores et al, Thalamocortical synchronization during induction and emergence from propofol-induced unconsciousness, Proc. Natl. Acad. Sci. USA 114 (2017)
Minor criticism:
The histological data are very impressive. In many cases, information on brain slices is indicated in small font, without a background. Therefore, it is very difficult to read, is it possible to change the font and / or add a contrasting background under the letters?
The presentation of a subject is systematic and comprehensive and analysis is proper. I am happy to recommend the manuscript for the publication after minor corrections mentioned above.
Round 2
Reviewer 1 Report
Thank you for your revision. Most of the comments that I raised are addressed. However, there are still issues that should be addressed before publication.
1. Firstly, why did the authors use a mixture of ketamine/diazepam instead of ketamine alone? It is important to consider the potential impact of diazepam use on the EEG and immunohistochemistry results. How do you think the use of diazepam may have affected the outcome of this study?
We have used the ketamine/diazepam mixture as one of the first anesthetics of choice for stereotaxic procedures and implantation of EEG and EMG electrodes for sleep recordings. In addition, only this ketamine anesthetic mixture is approved as a veterinary drug in place of ketamine/xylazine anesthesia in our country. The effects of ketamine/diazepam on local and global EEG microstructure during ketamine/diazepam-induced unconsciousness compared with propofol-induced unconsciousness are discussed in the first five paragraphs of the Discussion, especially the GABA agonist effects of diazepam in paragraph 5 (lines 158-161), and the last two paragraphs of the Discussion.
-> Main concern of the drug regimen used in this study is the authors used a mixture of two different drugs that since you described ketamine anesthesia, not ketamine + diazepam throughout the figures. Although diazepam is classified as a sedative clinically, however, it is evident that the addition of diazepam can intensify the effect of ketamine on hippocampal and cortical neurons compared to ketamine alone.
2. Page 15, line 410 “.. included only artifact-free recorded signals…”. How were artifact-free signals selected in this study? Were there any preprocessing steps before analysis?
No, there were no preprocessing steps before analysis. In this study, we used only signals of technically good quality from both brain structures for further analysis. Any signals containing artifacts were excluded.
-> Did you exclude the signals containing artifacts with a visual inspection? How can you ensure that the artifacts are excluded fully? Bandpass filtering or notch filter to eliminate line noise was not done in this study?
3. In the discussion section, the majority of the discussion focuses on reviewing previous literature. While it is important to consider the results in the context of the literature, more space should be devoted to discussing the results. For example, why does the alteration of interneuron expression/activity underlie different states of unconsciousness, even though anesthesia and sleep induce unconsciousness, which is indistinguishable from each other?
Our study shows that although anesthesia and sleep share many neurobiological features, they are distinct states in terms of local EEG microstructure and its underlying GABAergic and molecular substrate in different brain structures (distinct local neuronal networks) that are important for unconsciousness and EEG rhythm formation.
-> I can not see any improvement in the discussion section. Please revise this part accordingly.
